# Innovative Trends in Implant Dentistry Training and Education: A Narrative Review

**DOI:** 10.3390/jcm8101618

**Published:** 2019-10-04

**Authors:** Ana Santos Ferro, Ken Nicholson, Sreenivas Koka

**Affiliations:** 1Department of Oral Surgery, Malo Clinic, 1600-042 Lisbon, Portugal; anapestanaferro@gmail.com; 2Department of Periodontology, Lisbon Dental University, 1649-003 Lisbon, Portugal; 3ProfiVision Inc., Criagavon BT67 0QW, UK; ken.nicholson5@btinternet.com; 4Surgical Based Dentistry & Blended Learning, University of Central Lancashire, Preston PR1 2HE, UK; 5Advanced Prosthodontics, Loma Linda University School of Dentistry, Loma Linda, CA 92350, USA; 6Koka Dental Corporation, San Diego, CA 92111, USA; 7Advanced Prosthodontics, University of California Los Angeles School of Dentistry, Los Angeles, CA 90095, USA; 8Career Design in Dentistry, San Diego, CA 92111, USA

**Keywords:** implant dentistry education, blended learning, virtual reality, augmented reality, haptics, online learning, mixed reality

## Abstract

Background: The field of implant dentistry education is rapidly evolving as new technologies permit innovative methods to teach the fundamentals of implant dentistry. Methods: Literature from the fields of active learning, blended learning, augmented reality, artificial intelligence, haptics, and mixed reality were reviewed and combined with the experience and opinions of expert authors. Both positive and negative aspects of the learning methods are presented. Results and Conclusion: The fundamental objectives of teaching and learning remain unchanged, yet the opportunities to reach larger audiences and integrate their learning into active experiences are evolving due to the introduction of new teaching and learning methodologies. The ability to reach a global audience has never been more apparent. Nevertheless, as much as new technology can be alluring, each new method comes with unique limitations.

## 1. Introduction

Advances in technologies that are changing the way that dentistry is practised require a change in what is taught to dental students and dentists seeking professional education after formal university training has ended. In parallel, advances in technology offer exciting opportunities to improve the way that dentistry is taught, both as a result of new learning methodologies available and new learning methodologies being required for an evolving student population. The challenge is upon us as educators to address a fundamental choice. We can either choose to insist that traditional methods are the way forward, or we can embrace the opportunity presented by the learning styles of today’s students and the new capabilities afforded by innovative technologies.

Thinking back to traditional professional education, universities were knowledge-discoverers and knowledge-sharers; a student had to listen to the professor who would share her/his knowledge, and a student could only know what the professor was willing to share. Under the umbrella of “knowledge is power”, one element of Universities’ power was in their role as gatekeepers of knowledge sharing. In dentistry, the concept of problem-based learning introduced 15–20 years ago was the first shift away from students being limited by their professors’ knowledge [1]. Today, access to the world wide web offers the potential to access information about any topic; the world wide web has diluted the power of universities. A response of forward-looking universities has been to shift their attention from knowledge-discovery and knowledge-sharing as their *raison d’être* to knowledge-application and problem-solving. At Massachusetts Institute of Technology, their recently introduced primary value statement is “Ideas made to matter”—a thought-provoking statement in that it encapsulates discovery of new knowledge (Ideas), the application of knowledge (made) with a focus on issues of significance (to matter). The statement also feeds back on itself in that it demands that the ideas should be about things that matter so that one finds a problem of significance first and then one ideates and tests to solve the problem.

One challenge in implant dentistry education is that professional learners who wish to learn are still, too often, taught in traditional ways of knowledge sharing rather than involving students directly in the application of knowledge to solve, or better still prevent, clinical problems. In this paper, we share non-traditional methods that are already scientifically validated, and which open up new avenues for educators to have a greater impact and help our professional students provide a higher level of patient care.

Ultimately, our goal as educators today is no different than what it was a 10 or 50 years ago; to prepare our student learners to effectively, and compassionately and ethically provide patient-centred care that realizes individual patient aspirations for his/her care. In this manuscript, concepts such as action learning and blended learning are described, and a glimpse of the future is presented as artificial intelligence (AI), and machine learning will undoubtedly revolutionize educational methodology.

## 2. Action Learning

Action learning was originally developed by Revans in 1982 [2]. It is an approach in learning where people are committed to solving real-life problems by taking actions and thinking about it [2,3]. The student needs to be engaged and should take a personal and active role in all aspects of learning construction and problem-solving processes [3]. In dental education, action learning is commonly used since the major goal of dental education is a progression of knowledge acquirement, competencies, performances, and skills which will lead to clinical application [4,5]. Contrary to “Inert Knowledge”, where students are unable to apply skills and knowledge in a new situation, in dental education, a pro-active solution-driven thinking is mandatory since every patient to be treated is a unique challenge and will require a solution that meets their individual constellation of values, preferences, and beliefs [6]. Therefore, it is important to expend significant effort to create rich learning environments that involve students in learning activities in order to foster the application of what has been learned to new problems [2,3]. How is it possible to create this type of interactive environments in dental implant education?

Today, traditional theoretical lectures are being replaced by lectures with awe-inspiring images, dynamic videos, and interactive exercises. TED talks may be no longer than 18 min, and speakers are often invited to present for only 3, 5, or 9 min. They are short, capture attention, and, based upon the number of online views of TED presentations, highly effective. TED emphasizes the importance of meaningful presentation content that is focused on ideas and solutions—in essence, TED ensures that content is interesting. Boring lectures, long or short in duration, are easily substituted by cellular phones. 

How do we help our student to increase his/her level of competence, what new trends can help us to bring an attendee/student to be a better knower and develop into a doer?

Learning has become more challenging than ever, but the development of technology has led to the most engaging interactive exercises and fun learning. There are several online applications that can help the interaction between the speaker and the audience. 

### 2.1. Interactive Exercises and the Use of Online Applications

#### 2.1.1. Onsite Learning

Mentimeter, for example, is an application (app) one can use while giving a presentation, that allows for a quiz with several questions during the slideshow to be given to attendees who then answer. The app marks the time and accuracy of answers which allows the presenter to see if the students are acquiring the necessary knowledge. Voxvote is an app used at worldwide congresses that allows attendees to make anonymous questions to a moderator or speaker. It is very helpful because, for the attendee, since the questions remain anonymous, fewer are too afraid or shy to ask a question, and for the moderator, the pool of questions allows for screening and selection of the best questions that support his/her learning objectives. 

Sli.do is also used in conferences for the audience to submit questions, and once the Question and Answer session starts, the questions are displayed onscreen, and the speakers can address the most voted questions. The speaker can also submit a few of his own questions once the session gets underway. This will inspire the attendees to follow the lead and join with their own questions.

#### 2.1.2. Online Interactive Learning

Today, it is possible to have small working groups with experts from all over the world. Different time zones are no longer a barrier for exchanging knowledge and learning together. Social media have brought us modern study clubs through private groups in Whatsapp, Facebook, or even in Linkedin. People are invited to bring patient cases, challenges, and complications they are facing. The participants in the group help each other on clarifying the challenge and will help to create a solution and explore options and alternatives for action. Zygoma implant group, All-on-4 2018, Implantology/Oral Surgery, Esthetic Oral Surgery, Perio Freaks, FOR Emerging Leaders, are just some of these examples. Instastories (on Instagram) can be highly interactive since one can ask questions and conduct polls regarding any problem or challenge.

In addition, webinars and Facebook Live sessions allow for knowledge acquisition which can be done everywhere, even from one’s home, and can still be very interactive because of the opportunity to ask questions in the live question and answer sessions.

Many web platforms have online forums where dentists and students can learn from peers, participate in topical discussion, as well as share ideas and clinical cases. The distribution of information via these forums is vast, given the thousands of dentists and students worldwide who access them every day. 

These types of online interactive learning information cannot be used to obtain a degree but speak to the freedom of movement of content and the ability to access information from a variety of experts across the globe who could provide contemporary knowledge to the dental implant education environment [7].

### 2.2. Hands-on and Education Models

Worth affirms that learning Science is more than just gaining the facts and understanding of a particular topic [8]. This is where learning science through hands-on experiments becomes acceptable as an effective option, as it encourages students to experience and discover from observation or feelings [9].

Hands-on exercises in implant dental education are essential interactive exercises in which the practitioner will train and gain the ability to perform a certain intervention in a simulated environment.^7^ The teacher becomes a facilitator of a knowledge-sharing process. A wide range of different education hands-on models and exercises can be used.

#### 2.2.1. Standard Plastic Models 

Many standardized plastic models mimic patients’ real situations, being an excellent way of simulating implant placement. Single implants, partial cases, post-extraction, total rehabilitations, sinus lift, tilted implants, and other different scenarios have been reproduced in standardized implant models, achieving more realistic hands-on experience. Many models already present different types of bone quality, soft tissue, or even the Schneiderian membrane, and also have the possibility of being customized. Frasaco or BoneModels are two of the companies that produce these types of Education Models.

#### 2.2.2. Digital Models and Digital Planning

The introduction of Cone beam CT in dental offices and the existence of dental implant placement planning software has upgraded the possibility of diagnosis and planning of implants. Implants are planned, placed, and restored according to a chosen treatment plan and positioned related to anatomical and occlusal information. These plannings can be saved and shared with supervisors or peers, who can evaluate and give their opinion, making an excellent way of learning with real situations. DTX Studio, Blue Sky Plan, NeoGuide System, DIOnavi, and Simplant are some of the software available for digital implant planning, and some of them can be downloaded without any costs.

#### 2.2.3. Customized Printed Models

3D printing has become very popular in the dental field. It has a huge variety of applications—dental implant education being one of them. The Dicom files collected from a CBCT are converted into STL files enabling the fabrication of a patient’s bone model through 3D printing. The models are based on real patients, thus providing students with a more realistic preparation for real clinical situations [10]. These simulation models will contribute to a more interactive dental education and allow surgeons to do the procedure and overcome challenges before surgically implanting it into a patient. 

#### 2.2.4. Animal and Human Cadaver Models

Animal and human cadavers allow excellent training for students regarding the manipulation of soft and hard tissues. Flap design and elevation, osseous handling, drilling, and suture are very much similar to real-life tissues. These types of models are commonly used worldwide in many different interactive, hands-on sessions on implant education.

### 2.3. Navigation Hands-on

Dynamic navigation systems use a stereo vision computer triangulation setup to guide implant placement. It uses optical technologies to track the patient and the handpiece and to display images onto a monitor. The implant placement is guided at the proper depth, angulation, and crestal position according to the previous digital planning [11]. This type of hands-on allows the student to be corrected or guided through the intervention, allowing for self-improving skills with low error margin. 

### 2.4. Research Supporting Effectiveness of Action Learning and Hands-on/Interactive

According to Carin and Bass, ‘there are three major ways for people to learn about the world; discover things about the world from personal observations and experiences with the environment, acquire knowledge transmitted directly from other people or construct personal knowledge by transforming discovered and acquired knowledge in meaningful ways’ [11]. 

A study carried among twenty-two 4th-year students identifying the effectiveness of the hands-on experiment approach in science lessons has proven that hands-on experiments promote students’ learning and builds on their intrinsic motivation [12]. In a prospective study conducted with undergraduate medical students, the group trained according to a “best practice” model performed better in two skills of different complexities (nasogastric tube insertion; intravenous cannulation) than students trained with a traditional “see one, do one” teaching approach [13]. After a questionnaire on hands-on sessions on 3D printed models, students saw it as a great idea to provide additional learning and to be useful as a preparation for clinical courses [10]. In a survey administered to 372 undergraduate students who experienced both passive and active learning, the results showed that active instruction has more positive effects on student psychological and behavioral outcomes. Interestingly, a single active learning instructional session may be sufficient to yield significant and sustaining student learning outcomes [14].

Many different studies have been conducted about the effectiveness of action learning [15]. Acker-Hocevar’s major findings were that Action Learning helps leaders deal with ambiguity, build community, reinforces managerial, transformational, political and professional aspects of leadership, and promotes the application of previous academic learnings to real problems [16]. Butterfield believes that thought-provoking questioning is most effective in promoting learning as it has a long-term impact on improving questioning skills [17] whereas Choi states that Action Learning improved all eight coaching skills examined—relationship building, setting and communicating clear expectations, observational skills, analytical skills, listening skills, feedback skills, and creating a supportive environment [18].

### 2.5. Limitations

Creating action learning experiences is costly as more resources are often needed. These resources may be in the form of facilitators, equipment, and space, which are not needed when a single instructor gives a lecture in a lecture hall. Hence, the outcomes of the learning experience must be determined relative to the cost in order to assess whether the value to the participant and to the facilitators is warranted. Following on, action learning requires excellent facilitators who are well trained and highly engaged since the learning experience is so dependent on live and frequent engagement between participants and facilitators. 

In an action learning environment where participants are taking on new processes, there can be a sense of discomfort due to the unfamiliarity. Not all learners are comfortable revealing what they do not know to other participants who may be strangers. It is important, therefore, that learners be given the opportunity to reflect and share their learning experience, which allows the positive environment of action learning to be maintained. Often, the sharing of experiences reinforces the team learning environment.

## 3. Blended Learning

The term “Blended Learning” (BL) is becoming more commonly encountered in dental implant training and education. Nevertheless, there is confusion regarding what constitutes Blended Learning and how it is best defined. The plethora of available definitions led Oliver and Trigwell [19] to suggest that the term “blended” should be “abandoned or, at least radically reconceived.” They continue that even the term “learning” should be abandoned on the basis that current definitions relate more to the perspective of the course designer than the learning needs of the student. 

Nevertheless, when considering some of the definitions in the literature, clearly there is much common ground. Rowe et al. describe blended learning as the systematic integration of online and face-to-face engagement to support and enhance meaningful interaction between students, teachers, and resources [20]. They emphasize that blended learning is highly context-dependent. Blended learning concepts that work within one domain may not have value in another domain. Distilling the available definitions of BL, and taking into consideration the work of Knowles [21], a framework can be assembled to help determine what BL could or should be. The common denominator is that a BL course should be learner-driven and developed to take into consideration the needs of the individual learner. Following on, Means et al. [22], in their meta-analysis of the empirical literature on the effectiveness of online and blended learning considered a course “Blended” only when a minimum of 25% of course content is delivered online. All forms of media and technology should be considered in the design of a program and their use optimized to meet the needs of the learner.

Presently, the majority of implant education and training occurs after a dentist has been awarded his/her primary degree. As such, adult learners are the core group of learners in implant dentistry. The attributes and needs of the adult learner as discussed by Knowles [21], are presented below:Have a specific purpose in mindAre voluntary participants in learningRequire meaning and relevanceRequire active involvement and learningNeed clear goals and objectivesNeed feedbackNeed to be reflective

### 3.1. Blended Learning Models

A variety of Blended Learning models that combine a mix of synchronous and asynchronous formats are presented below.

#### 3.1.1. Synchronous Physical Formats

Instructor-led classrooms and lecturesHands-on laboratory and workshop sessionsHands-on clinical training

#### 3.1.2. Synchronous Externally-Directed Learning Formats

Virtual classroomsWeb seminars (webinars) and broadcasts/podcastsInstant messagingE-meetings

#### 3.1.3. Online Live E-Learning Formats That Are Self-Paced and Asynchronous

Documents and web pagesWeb/computer-based training modulesAssessments/tests/surveysRecorded live eventsOnline streaming and discussion foraSimulated experiencesJob aids and electronic performance support systems (EPSS)

From the perspective of implant dentistry education, a practitioner aiming to offer implant therapy to his or her patients will need to acquire the necessary skills to do so, and the knowledge to underpin those skills. Blended Learning provides the opportunity to use internet-based technology to facilitate the delivery of the knowledge component of a program and face-to-face mentoring to deliver the skills component.

When investigating the world’s first Massive Open Online Course (MOOC) in implant dentistry, Lan et al. demonstrated how internet-facilitated knowledge delivery was well suited to many of the self-regulated learning strategies they identified as part of their research [23]. The authors continued that implant dentistry is a clinical discipline and cannot be learned solely online. Perhaps the term “Balanced Learning” should be introduced, especially in relation to implant dentistry, to ensure the learner acquires appropriate knowledge, application, and problem-solving to result in the provision of competent clinical care. It is clear, however, that the implicit constraints of time, geography, and format of the physical classroom can be overcome with the use of technology. Unfortunately, uptake and implementation of BL programs within mainstream dental implant education have been slow, even though numerous studies have indicated it provides superior outcomes when compared with more traditional teaching methods [24,25,26,27].

### 3.2. Artificial Intelligence (AI) and Machine Learning (ML)

The introduction of Google Glass in February 2013 revived interest in artificial intelligence (AI). Since 2013 the growth of research into AI and use of AI has been developing rapidly. The use of technology in BL programs brings with it the opportunity to incorporate significant advances in computing power. The combination of increased computing power and volume of data collected by organizations such as Google, Amazon, Microsoft, and IBM have led to what has been heralded as the “4th industrial revolution”. According to the Oxford dictionary, artificial intelligence can be described as the theory and development of computer systems able to perform tasks normally requiring human intelligence, such as visual perception, speech recognition, decision-making, and translation between languages.

The foundation of AI is machine learning (ML) and is underpinned by statistics and grouping and classifying data. More complex AI takes the form of neural networks and the quest to mimic human intelligence. At a fundamental level, neural network systems take large amounts of data (the input layer) and apply numerous algorithms to weight the data (the hidden layer) in order to provide an output. Complex neural networks lead to what has been termed “Deep Learning”. 

### 3.3. Augmented Reality (AR), Virtual Reality (VR) and Mixed Reality (MR)

Unsurprisingly, AI is impacting education. New technology allows us to experience an enhanced sensory perception of our environment using AR in which computer-generated virtual content is superimposed on real structures. Through AR, a user feels as though she/he is connected in an enhanced environment. This is in contrast to VR, which brings together a combination of multiple technologies allowing users to interact with virtual entities in real-time. The combination of AR and VR has been termed Mixed Reality (MR). At the time of writing, the most recent advance with MR is exhibited by the Microsoft Hololens 2 (Microsoft Corp. Redmond, WA, USA) with its primary use being education. Kwon et al. state that [28] “AR technology has been shown to substantially improve a wide range of procedures” including dental implant placement. This is achieved by means of an AR surgery navigation system incorporating a head-mounted display. The authors emphasize that having a head-mounted display is safer than averting one’s gaze to a monitor while performing surgery.

Cost savings and overcoming a shortage of teachers and supporting faculty are often cited as reasons for implementing the use of technology in dental education. Elby et al. [29] describe the need for virtual reality simulators in dental education. They discuss the use of high-fidelity robotic patient models that cough, tongue thrust, shake the neck and secrete saliva in aiding the teaching of medical emergency management in the dental setting. However, literature specifically addressing the use of AR/VR in dental implant education is scarce. Xiaojun et al. [30] introduced a modular software package named CAPPOIS (Computer Assisted Preoperative Planning for Oral Implant Surgery) used in conjunction with a VR enabled haptic feedback system for teaching implant treatment planning and subsequent surgery. The system provides the operator with tactile feedback during virtual surgery based upon the treatment plan and relevant anatomy at the surgical site. Using such a system, the experienced operator can practice the surgery in a virtual environment prior to actual treatment on the patient. An inexperienced operator can plan the procedure and carry out their procedure in a mixed reality environment, guided by both visual and haptic feedback. The authenticity of the learning experience will be determined by the number and type of components built into the MR space. 

As the paper is considered for publication, the features of MR offered by the Hololens 2, referred to earlier, are being explored, evolved, and used to aid surgical treatment and student training. Figure 1, Figure 2, and Figure 3 demonstrate how powerful and useful AI, AR, VR, and MR may become.

### 3.4. Limitations

A significant limitation to any AR/MR/VR system and how quickly adopted is the quality of user experience design. To construct sufficiently “real” environments requires expensive and sometimes technically challenging interface design, especially with regard to optics and holographics. Following on, creating an acceptable interface for the user from a realistic view remains insufficient for mainstream acceptance and usage as both user convenience and user control need to be addressed. Here, “user convenience” relates to the availability of choices and accessibility relative to the cost/functionality ratio, where the cost to the user of using the system is not so high as to prevent acceptance. This is especially true for low-cost applications where the cost of creating the system prevents uptake. In addition, “user control” relates to the ability of the user to control the pertinent devices with sufficient sensitivity and real feel, especially with movement control.

Another major issue, and potentially important limitation, revolves around data privacy and user privacy. For example, web platforms are a valuable source of shared knowledge but must comply with relevant laws and regulations that govern oversight of big data management. These laws and regulations seek to respect ethical, legal, and social standards. In health care, protecting patient privacy and patients’ rights is paramount, and the principles of informed consent must be followed to allow patients the ability to control how their data is shared [31].

## 4. Summary

Education in implant dentistry will evolve quickly over the next decade as technologies already being used in other industries are incorporated into new and innovative learning models. The merging of technological innovations culminating in “digital dentistry” make the “digital education of digital dentistry” inevitable. Going forward, instead of traditional models of education being used to achieve all educational objectives, now traditional formats that will be ineffective with today’s learner will be replaced, where appropriate, with online education, AR, ML, VR, and MR. Of course, an important need in education will still be addressed through one-on-one mentoring and lecture format teaching models. However, the restraints of these models will be released through innovative use of new technologies that allow for different, deeper, local and global, near and distant educational experiences.

## Figures and Tables

**Figure 1 jcm-08-01618-f001:**
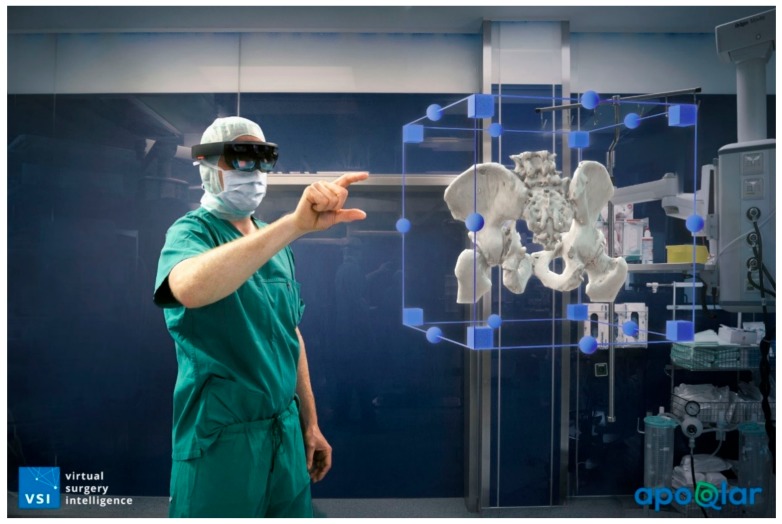
The image indicates how the operator can visualize a part of the patient’s anatomy. If 3D imaging is used as part of case assessment, an operator can be presented with a hologram of the patient’s anatomy that can be manipulated with simple hand gestures.

**Figure 2 jcm-08-01618-f002:**
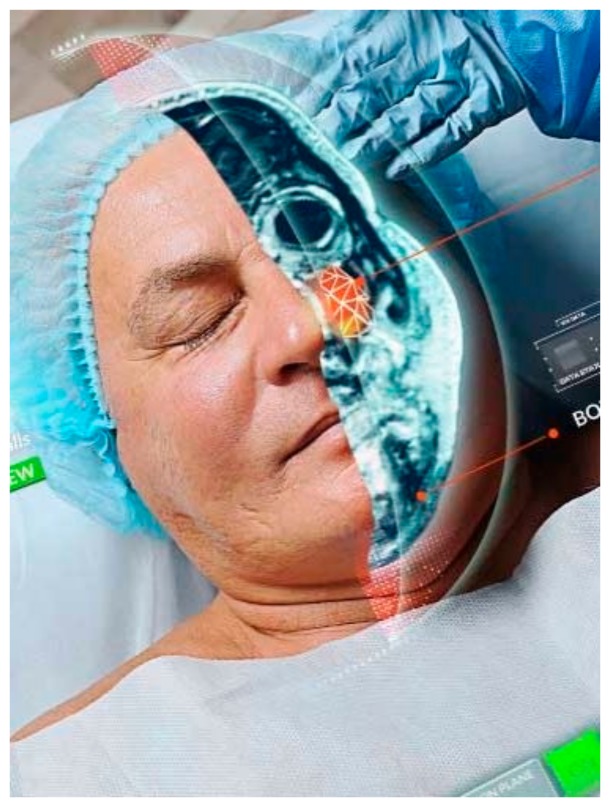
Diagnostic imaging can be overlaid on the patient prior to and during surgery. Many features can be added to the virtual MR world to allow the operator to access information in many domains at many levels of detail with a simple hand gesture.

**Figure 3 jcm-08-01618-f003:**
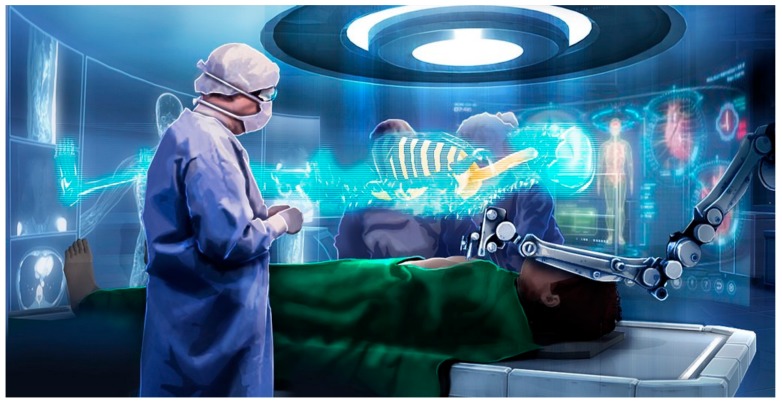
Replacement of a live patient by a virtual patient using MR. The risk to a live patient is reduced. The MR world, which is a simulation of the real world, can be enhanced using AI to meet the needs of the individual learner. Thus, specific learning outcomes for the adult learner can be more readily achieved.

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
