# Peer review of "Innovative Trends in Implant Dentistry Training and Education: A Narrative Review"

_jcm, 2019, doi:10.3390/jcm8101618_

Round 1

Reviewer 1 Report

In the present review, the innovative methods in postgraduate education in implant dentistry are summarized. The topic of the review is interesting and the review is timely because the importance of new technologies in education will be growing in the future. Some points, which need to be considered by Authors are as follows:

The major weakness of this review is the absence of critical analysis of the innovative learning methods. The manuscript describes only the advantages of new learning methods but ignores their limitations. A critical analysis is a major feature of a scientific paper. The type of review should be mentioned. Is it narrative or systematic Review? How the literature search was performed? The review has only 30 citations, which is rather low for such kind of manuscript. The abstract is too short and consists only of 4 rows. There are no subsections like objective, methods, and conclusions. Some sentences are written as for the boulevard press, for example, “The beautiful thing regarding technology is that…”, “An example of AI in our daily life is…” etc. The standards of scientific writing should be kept for the whole text. There are many inconsistencies in the text formatting throughout the manuscript. The format of the reference list is inconsistent, Refs 1-13 vs 14-30.

Author Response

Thank you to reviewer #1 for his/her helpful comments to improve our manuscript.

1.  We have expanded the abstract and added structured sections as requested.

2.  The scientific writing level has been improved to be more consistent throughout the manuscript and to remove colloquialisms.

3.  The manuscript has been defined as a narrative review.

4.  Text formatting has been improved and references are now consistently formatted.

5.  Each major section has been extended to include information regarding the limitations of the learning methods discussed.

We hope our revisions meet with the reviewer's approval and thank you again for your thoughtful recommendations.

Reviewer 2 Report

The review is well written and explains the novel learning methods (VR, AR, MR, AI) and how they are applied in implant dentistry.

It is important topic to discuss as it is rapidly developing.

The manuscript is well structured, so the reader can follow easily different aspect of learning.

However, authors didn’t include clear aim and the methodology.

Authors presented the advantages of different learning methods. The article will benefit if authors will include also critical approach. 

One of the example is learning via web platforms.

While such platforms are a valuable source of shared knowledge, they should also comply with relevant laws and regulations (e.g. at the EU level with GDPR and national regulations), and respect ethical values and follow ethical standards including research ethics. Particularly, patients' rights should be respected, including privacy and informed consent.

Open source science is not free from bias, even unintentional bias or discrimination. Researchers should respect fundamental human rights and follow principles of research ethics.

The ethical, legal and social implications of health-related big data raise novel challenges. The ethical, legal and social implications of health-related big data raise novel challenges.

I would suggest to consider including legal and ethical aspects. To help the authors to find further information below is a link, that further information can be found.

https://ec.europa.eu/health/home_en

Author Response

We thank reviewer #2 for his/her insightful and helpful comments.  We have revised the manuscript accordingly.

The abstract has been revised to provide the reader with additional clarity about the structure and methods of the pape. The title has been revised to state that this paper is a narrative review. A critical approach has been included with new sections on Limitations of Action Learning and Blended Learning/VR/MR/AI/AR. The issue of big data and privacy and the ethical/social issues has been included in the limitations section for Blended Learning.  The web site reference offered by the reviewer has been added to the reference section.

Thank you again for your very helpful review comments.

Round 2

Reviewer 1 Report

All criticism points were considered in the revised version.